# Impact of COVID-19 Pandemic on Well-Being, Social Relationships and Academic Performance in a Sample of University Freshmen: A Propensity Score Match Evaluation Pre- and Post-Pandemic

**DOI:** 10.3390/ijerph20156485

**Published:** 2023-07-31

**Authors:** Chiara Buizza, Clarissa Ferrari, Giulio Sbravati, Jessica Dagani, Herald Cela, Giuseppe Rainieri, Alberto Ghilardi

**Affiliations:** 1Department of Clinical and Experimental Sciences, University of Brescia, Viale Europa 11, 25123 Brescia, Italy; giulio.sbravati@unibs.it (G.S.); jessica.dagani@unibs.it (J.D.); herald.cela@uni-graz.at (H.C.); giuseppe.rainieri@unibs.it (G.R.); alberto.ghilardi@unibs.it (A.G.); 2Research and Clinical Trials Office, Fondazione Poliambulanza Istituto Ospedaliero, Via Bissolati 57, 25124 Brescia, Italy; clarissa.ferrari@poliambulanza.it; 3Department of Psychology, University of Graz, Universitätsplatz 2, 8010 Graz, Austria

**Keywords:** university freshmen, mental health, psychological distress, COVID-19 pandemic impact, social relationships, academic performance, coping

## Abstract

The COVID-19 pandemic has impacted freshmen, compromising their mental health, lifestyles, and academic performance. There are few studies that have investigated changes in the health status and lifestyles of freshmen before and after the pandemic. The aims of this study were: (1) to carry out a pre–post-COVID-19 pandemic comparison between two freshmen samples, in order to detect differences in their socio-demographic characteristics and in some clinical variables; (2) to assess the effect of the COVID-19 pandemic on the social and academic lives of the second sample of freshmen. The samples recruited in 2019 and 2022, matched by propensity score procedure (N = 553), were mostly female (57.3% vs. 55.3%); the mean age was 22.9 and 20.9 years, respectively. The freshmen recruited after the pandemic had less psychological distress and substance use than freshmen recruited before the pandemic. Seventy-eight percent of the freshmen stated that the pandemic had an impact on their social relationships. This effect was greater for females and Italian students. Forty-seven percent reported that the pandemic has worsened their academic performance, while 60% stated that pandemic has improved their grades. The results of this study can provide valuable insights into the impact of the pandemic on freshmen, in order to implement interventions to mitigate the consequences of the pandemic in some subgroups of this target population.

## 1. Introduction

The COVID-19 pandemic has brought about significant concerns regarding the mental health of university students. Beyond the physical health risks, the pandemic has emerged as a major stressor that has potentially compromised the mental health of students, leading to lifestyle changes and potential consequences for their overall well-being, levels of anxiety, academic performance, and self-efficacy [1]. Students who showed a higher level of anxiety expressed more negative emotions and perceived themselves with less academic self-efficacy. The pandemic, together with illness and the death of a relative/friend due to COVID-19, increased anxiety and jeopardized the perception of academic self-efficacy [2].

Previous research has highlighted the first year of college as a critical period characterized by heightened vulnerability to psychological distress and its subsequent impact on academic performance [3,4]. There are several factors that make this population vulnerable: freshmen can struggle to orient themselves in college, which requires different time and study management than high school; they must learn to self-regulate their knowledge, using different cognitive strategies; they often live off-site, away from their family, and it can take time to make friends, so they can live for a long time without social support; finally, students can enroll in university without a real intrinsic motivation, but following pressure from others (e.g., parents, etc.). All these factors, together with the changes in social roles and in the re-definition of identity that occurs in this stage of life [5], can represent important stressors. In fact, the transition from high school to university occurs during emerging adulthood. This coming together of multiple challenging development tasks at the same time may be stressful for some students. The COVID-19 pandemic, and above all the health measures implemented to deal with it, may have been an additional stress factor contributing to the difficult adaptation of freshmen to academic life. The Global Survey Report of the International Association Universities (IAU) on the “Impact of COVID-19 on higher education around the world” showed the important degree of stress and constraint experienced by higher education institutions during the pandemic [6]. Almost all institutions are affected by the COVID-19 pandemic and the pandemic has affected all institutional activities. With the emergence of the COVID-19 pandemic, university students have faced several challenges during their transition to higher education in the midst of a global health crisis. The sudden closure of universities and campuses, coupled with the abrupt shift to remote learning formats, has had a profound impact on the social and academic lives of university freshmen. The biggest change was the distance learning, which influenced the academic performance and the emotional factors. Universities had to quickly adapt to this new teaching methodology. Many studies on this topic have appeared in the literature, evaluating students’ satisfaction with remote teaching and showing several degrees of appreciation [7]. The IAU Report showed that remote activities were possible to some extent, but could have had negative effects on the quality of activities, also increasing the inequality of learning opportunities [6]. Only in the future will we be able to evaluate the real effects of e-learning [8]. With respect to the emotional aspects, the loss of face-to-face interactions with peers, faculty, and support services has contributed to heightened feelings of loneliness and social isolation [9,10]. Not attending face-to-face lessons made the students worried, nervous, and anxious [11].

These disruptions may have exacerbated the challenges already faced by university freshmen as they adapt to a new academic environment and establish new social networks. Furthermore, the uncertainty surrounding the duration and long-term consequences of the pandemic added an additional layer of stress for university freshmen. Concerns about the impact of COVID-19 on academic progress, future career prospects, and financial stability contributed to heightened anxiety and decreased psychological well-being [12,13]. Recognizing the specific challenges faced by freshmen is of utmost importance due to their heightened susceptibility to psychological distress, which has been associated with lower academic performance and an increased risk of dropout [14,15].

Despite the pandemic being a major stressor that has compromised the mental health of university freshmen and changed their lifestyles, leading to significant consequences for their well-being and academic performance, there have been few studies investigating changes in the health status and lifestyles of university freshmen before and after the COVID-19 pandemic. This is primarily due to the difficulty in gathering large prospective data samples from anonymous surveys. To overcome this limitation, we have adopted the propensity score matching technique, applied to samples of freshmen recruited through two different web surveys conducted before (2019) and after (2022) the COVID-19 pandemic. Although the World Health Organization (WHO) officially declared the end of the COVID-19 pandemic on 5 May 2023, in many countries, the emergency ended in 2022. In Italy, the closure of the state of health emergency declared on 31 January 2020 to counter the spread of the COVID-19 pandemic took place on 31 March 2022 [16]. For this reason, we considered it possible to talk about a post-pandemic timeframe in our research, considering a sample recruited from May to June 2022, when the state of emergency in Italy was over.

With this study, we wanted to fill the gap existing in the current international literature on the impact that COVID-19 has had on freshmen. Even with regard to Italian studies, this is still an unanswered topic. The published studies are almost always on university students and not on freshmen and have always been conducted during the health emergency, before the declaration of the end of the state of emergency issued by the Italian government [7,17]. To our knowledge, there is only one Italian study on freshmen that evaluated the role played by emotional processing and differentiation of self for psychological well-being during the pandemic, but this study was also conducted during the health emergency and therefore did not assess the impact of the COVID-19 pandemic on freshmen [18].

The aims of this study were: (a) to carry out a pre–post-COVID-19 comparison between two samples of freshmen, in order to detect possible differences in their socio-demographic characteristics and in some clinical variables such as well-being, mental distress, substance use, and suicide risk; (b) to assess the effect of the COVID-19 pandemic on the social and academic lives of the second sample of freshmen.

## 2. Materials and Methods

### 2.1. Study Design

The present study took place in a University in Northern Italy, composed of about 15,000 students. The courses of study that the university offers are grouped into 4 areas: medicine, engineering, economics, and law. The study involved two samples of freshmen: the first was recruited from May to June 2019 and the second from May to June 2022. The study was aimed at all freshmen attending university, and the participation was voluntary. For this reason, there were no inclusion or exclusion criteria. The freshmen received a first email asking them to participate in a web-survey, together with the link to access the survey and a complete description of the study. Through the web-link, freshmen were asked to confirm their informed consent to participate. The web-survey was created with LimeSurvey (www.limesurvey.org, accessed on 1 May 2019 and on 1 May 2022), an online survey tool that allows for completely anonymous data collection. The survey was implemented following the guidelines proposed by Pealer and Weiler [19]. To increase response rate, we used some of the strategies proposed by Edwards et al. [20] such as using user-friendly questions, choosing close-ended options for answers, and sending reminders.

### 2.2. Instruments

The tools listed below were administered to the two samples. Appendix A shows the detailed description of the tools.

The General Health Questionnaire (GHQ-12) is composed of 12 items that assess psychophysical well-being [21]. Each item scores from 0 (much less than usual) to 3 (more than usual). A score of 0 was assigned to the first two low-stress alternatives and a score of 1 was given to the two high-stress alternatives. The maximum score was 12, with a cut-off point > 3 indicating psychological distress. The GHQ-12 proved to be a reliable instrument, as indicated by a Cronbach’s alpha of 0.81 [22].

The University Stress Scale (USS) is composed of 21 items that capture the cognitive appraisal of demands across the range of environmental stressors experienced by students [23]. Students are asked to rate on a 4-point Likert scale, ranging from 0 (not at all) to 3 (constantly). The total score ranges from 0 to 63: the higher the score, the higher the perceived stress level (cut-off point ≥ 13 is predictive of mental distress). The USS proved to be a reliable instrument as indicated by a Cronbach’s alpha of 0.83 [24].

The University Connectedness Scale (UCS) is composed of 18 items that assess the degree of support and membership perceived by students with respect to their university [25]. Students are asked to rate on a 7-point Likert scale, ranging from 1 (not at all) to 7 (all the time). The total score ranges from 18 to 126, where the higher the score, the higher the student’s perception of belonging and support within their own university. The UCS has a strong internal consistency with a Cronbach’s alpha of 0.88 [26].

A modified version of World Health Organization-ASSIST v3.0, which is a questionnaire, based on the self-report adaptation of Barreto et al. [27], to detect harmful and hazardous drug use [28]. It contains 8 questions about 10 substance categories: tobacco, alcohol, marijuana, cocaine, methamphetamine/amphetamine type stimulants, inhalants, sedatives, hallucinogens, opioids, and other drugs. Each response corresponds to a score, ranging from 0 to 6, with the total score summation ranging from 0 to 39 for each substance. Total scores between 0 and 3 (0–10 for alcohol) are considered “low risk” (occasional or non-harmful use), 4 and 26 (11–26 for alcohol) indicate “moderate risk” (more regular use or harmful/hazardous use), and scores higher than 26 indicate “high risk” (frequent high-risk use or suggestive of dependence). The ability of the ASSIST to classify patients based on degree of drug use has been extensively validated [29,30]. Cronbach’s alpha values range from 0.71 to 0.90 [27].

The Brief COPE Inventory [31] is composed of 28 items grouped into 14 sub-scales that represent modalities for facing stress conditions: self-distraction, active coping, denial, substance use, use of emotional support, use of instrumental support, behavioral disengagement, venting, positive reframing, planning, humor, acceptance, religion, and self-blame. Each item scores from 1 (I haven’t been doing this at all) to 4 (I’ve been doing this a lot) and each scale ranges from 2 to 8: a high score is related to a greater ability to cope. The Brief COPE proved a reliability for each scale ranging between 0.50 and 0.90 [31].

The P4 Screener is a brief tool to assess potential suicide risk, which includes a pre-screening question about thoughts of self-harming [32]. If a positive answer is given to this pre-screening question, there are subsequent questions on the “4 Ps”: Past suicide attempts, Plan, Probability of completing suicide, and Preventive factors. Potential suicide risk is classified as minimal, lower, or higher. There is considered to be a ‘minimal’ risk when there is no past history, no suicide plan, and a “not at all likely” probability of an attempt. ‘Lower’ risk refers to respondents who indicated a plan and/or past history but responded “not at all likely” to the question regarding probability and noted there were factors preventing them from taking action. ‘Higher’ risk respondents are those who reported the probability of a suicide attempt as being either “somewhat likely” or “very likely” and/or reported there were no factors preventing them from taking action. If respondents give a negative answer to the pre-screening question, they are classed in the “did not trigger” category of risk.

The freshmen recruited in the second survey also completed the following instruments.

The Academic Motivation Scale (AMS) is a questionnaire developed on the basis of the Self-Determination Theory [33]. For this study, we used the adapted form developed by Biasi et al. [34] to answer the question: Why are you attending the degree program you are enrolled in? The AMS is composed of 20 items rated on an 11-point Likert scale ranging from 0 (not at all true) to 10 (completely true). The items are grouped into 5 sub-scales: lack of motivation, external regulation, introjected regulation, identified regulation, and intrinsic regulation. The total score ranges from 0 to 40, where a higher score corresponds to a greater adherence to the motivational construct of the single sub-scale. The AMS showed good psychometric properties with Cronbach’s alpha values ranging from 0.73 to 0.91 [35].

The Perceived School Self-Efficacy Scale (PSSES) is composed of 9 items and structured on a 5-point Likert scale ranging from 1 (not capable at all) to 5 (fully capable). For this study, we used the reduced and adapted form developed by Biasi et al. [34]. The focus of this scale is the perception that students have about their ability to regulate and focus on the studying process. The total score ranges from 9 to 45, where a higher score corresponds to a higher level of self-efficacy perceived by the student in terms of study and academic skills. The PSSES proved to be a reliable instrument with Cronbach’s alpha values ranging from 0.83 to 0.87.

The Self-Regulated Knowledge Scale-University (SRKS-U) is a questionnaire developed on the basis of Pintrich’s theory of self-regulated knowledge that assesses the frequency with which students implement different cognitive strategies [36]. The SRKS-U consists of 15 items rated on a 5-point Likert scale ranging from 1 (never) to 5 (always or nearly always). The SRKS-U is composed of 5 sub-scales evaluating the use of predefined cognitive processes: knowledge extraction, knowledge networking, knowledge practice, knowledge critique, and knowledge monitoring. The score of each subscale ranges from 3 to 15, where a higher score corresponds to a greater use of that cognitive strategy. The SRKS-U proved to be a reliable instrument with Cronbach’s alpha values ranging from 0.70 to 0.80 [37].

The Shortened Achievement Emotion Questionnaire (AEQ-S) [38] is a condensed version of the original Achievement Emotions Questionnaire, designed to assess college students’ achievement emotions [39]. In this study, we used the sub-scale “Learning-related Emotions” that measures students’ enjoyment, hope, pride, anger, anxiety, shame, hopelessness, and boredom in situations of studying. This sub-scale consists of 32 items rated on a 5-point Likert scale ranging from 1 (strongly disagree) to 5 (strongly agree). The sub-scale is computed by summing the items and taking the mean (no cut-off). Cronbach’s alpha values range from 0.75 to 0.93.

The South Oaks Gambling Screen (SOGS) is a screening tool to assess the presence of a gambling addiction [40]. It consists of 20 elements and includes 3 previous items that do not count towards the total score and are used to evaluate the type of game or bet, the maximum amount staked, and whether they are close to other people with gambling issues. The response options for items are dichotomous (“yes” or “no”). Scores on the SOGS are determined by scoring one point for each question that shows the “at risk” response indicated and adding the total points. The maximum score is 20 and a cut-off score of ≥5 indicates that the respondent is a Probable Pathological Gambler (PPG). The SOGS has a strong internal consistency with a Cronbach’s alpha of 0.97.

Furthermore, all freshmen were requested to fill out an assessment form, which provided information regarding their socio-demographic and academic characteristics.

For the evaluation of the effect of pandemic on social relationship and on academic life, ad hoc questions were set for the second (post-COVID-19 pandemic) survey. In detail, freshmen were asked to reply to the following three questions (one regarding social life and two regarding academic life): (i) Did the pandemic impact your social relationships? (ii) Did the pandemic improve your academic grade? (iii) Did the pandemic worsen your academic performance? Grades and academic performance were assessed at the end of the first semester. The responses were set on a 5-point Likert scale (strongly disagree; disagree; moderately agree; agree; strongly agree) and were then dichotomized as ‘no vs. yes’ responses by merging ‘strongly disagree + disagree’ vs. ‘moderately agree + agree + strongly agree’).

### 2.3. Statistical Analysis

Categorical data are presented as absolute (n) and percentage values (%). Differences between the two survey groups were tested using the Pearson’s Chi-squared test (or Fisher’s exact test when appropriate). Continuous variable distributions were described by mean, median, and standard deviation (SD). Differences between continuous variables were analyzed using a *t*-test or corresponding non-parametric Mann–Whitney U-test for comparing two groups of Gaussian or non-Gaussian distributed variables respectively.

Propensity score was calculated using a logistic regression model. Potential confounding variables/predictors were chosen in agreement with all available socio-demographic variables in common between the two surveys. Those predictors comprised: age, gender, nationality [Italian vs. other], marital status [single vs. in relationship], university status [student vs. working student], living status [in town vs. out-town]. We set the matching tolerance to 0.2. Matching was performed in a 1:1 ratio of nearest neighbor without replacements. We excluded freshmen with incomplete information on predictors.

To assess the association of the socio-demographic and clinical variables with the outcomes regarding the effect of the pandemic on personal life (one outcome assessing the perception of the pandemic’s impact on social relationships) and on academic life (two outcomes assessing the perception of the pandemic’s impact on examination grades and on the overall academic performance, respectively), univariate and multivariable logistic regression models were applied. For the choice of the best predictors in the multivariable models, the stepwise method was applied.

Data analysis was performed using the R software v.4.2.2. For the propensity score, the MatchIt package was used, while for the stepwise method, the step AIC function of package MASS was used.

## 3. Results

### 3.1. Comparison between Two Freshmen Samples

The pre-COVID-19 sample collected in 2019 was composed of 553 freshmen while the post-COVID-19 sample collected in 2022 counted 721 freshmen. Considering the original survey collected data (i.e., before the propensity score matching), the differences between the two samples concerned age, marital status, place of residence, working student status, GHQ-12 total score, some scales of the Brief COPE, and the use of some substances categories included in the ASSIST (Appendix A).

Table 1 shows the differences between the two samples after the propensity score matching for socio-demographic variables (age, gender, nationality, marital status, university status, living status). Matching for age failed due to a severe imbalance between the two surveys for this variable; for this reason, all the subsequent analyses were then adjusted for age. The two samples differed by the GHQ-12 total score: freshmen recruited after the COVID-19 pandemic had lower scores than freshmen recruited before the pandemic, even though the score was indicative of psychological distress in both samples (cut-off > 3).

With respect to coping strategies, freshmen recruited in 2022 showed lower scores, compared to freshmen recruited before the pandemic, in: planning, i.e., the attitude to identify the most suitable strategies to resolve the situation; acceptance, that is, the ability to live with difficulties; and self-blame, that is, the attitude to attribute the occurrence of events to oneself. Freshmen recruited after the pandemic showed instead higher scores, compared to pre-pandemic freshmen, in: active coping, indicative of a greater attitude to focus on the situation and to develop strategies to improve it; and substance use (two Brief COPE questions on using alcohol and drugs to cope with stress and feel better). In contrast to the latter data, the post-COVID sample reported lower scores, indicative of lower use, in some substances evaluated through the ASSIST: tobacco, alcohol, marijuana, and cocaine. As regards the other substances, there were no differences between the two groups.

### 3.2. Effect of the COVID-19 Pandemic on Social Relationships

Considering the post-pandemic survey, the majority of freshmen (78%) declared that the pandemic has had an impact on their social relationships (Appendix A). The association between the socio-demographic and clinical variables and the perception of the pandemic’s impact on social relationships in the sample of freshmen recruited after the pandemic was evaluated by univariate logistic models. The significant variables were: gender (*p =* 0.021), nationality (*p =* 0.004), USS total score (*p =* 0.002), emotional support (*p =* 0.005), instrumental support (*p =* 0.003), venting (*p =* 0.030), lack of motivation (*p =* 0.042), external regulation (*p =* 0.020), pride (*p =* 0.022), shame (*p =* 0.014), and hopelessness (*p =* 0.005) (Table 2). By applying a multiple (multivariable) logistic model with all these significant factors as independent variables, and by applying the step AIC variable selection procedure for finding the best fitting, the best obtained model is shown in Table 2. The results showed that being female, compared to being male, increased the probability by 69% (OR = 1.69, 95%CI: 1.09, 2.62) of saying that the pandemic has had an effect on social relationships compared to saying that it has had no effect. The probability of saying that the pandemic has had an effect on social relationships, compared to saying that it has had no effect, was 76% lower among non-Italian freshmen than among Italian ones. This means that Italian students have suffered more negative effects on social relationships due to the COVID-19 pandemic. As the USS increased by 10 points, the probability of saying that the pandemic has had an effect on social life increased by 40% (OR = 1.04, 95%CI: 1.01, 1.07). Having an external regulation increased the probability of saying that the pandemic had an effect on social relationships compared to saying that it had no effect (OR = 1.035, 95%CI: 1.002, 1.06). Similarly, using instrumental support increased the probability of saying that the pandemic had an effect on social relationships (OR = 1.17, 95%CI: 1.03, 1.34).

### 3.3. Effect of the COVID-19 Pandemic on Academic Performance

The effect of the pandemic on academic life was evaluated by considering both the effects on the examination grades and on the overall perceived academic performance. Regarding the examination grade, 60% of freshmen declared that the pandemic contributed to improving their examination grade (Appendix A). The socio-demographic and clinical variables significantly associated with the perceived impact of the pandemic on examination grade were: active coping (*p =* 0.043), humor (*p =* 0.049), venting (*p =* 0.011), sedatives (*p =* 0.007), knowledge monitoring (*p =* 0.020), intrinsic regulation (*p =* 0.028), IAUQ total score (*p =* 0.018), and SOGS total score (*p =* 0.032) (Table 3). Applying the multivariable logistic model and selecting the most predictive variables using the stepwise procedure, the best estimated model included venting, sedatives, intrinsic regulation, and SOGS total score (Table 3). The results showed that a unit increase in venting and in sedatives score increased the probability by 15% (OR = 1.15, 95%CI: 1.03, 1.29; 95%CI: 1.05, 1.31, respectively) of saying that the pandemic has improved the examination grade compared to saying that it has had no effect. Similarly, a unit increase in SOGT total score increased the probability by 24% of saying that the pandemic has improved the examination grade. Differently, there was an inverse association between intrinsic regulation and perception of improvement in examination grades: a higher score in intrinsic regulation was associated with a reduction of the perception of the improvement in examination grade of 2% (OR = 0.98, 95%CI: 0.96, 0.99).

Finally, when freshmen were asked about their perception of the pandemic’s impact on their overall academic performance, their response was quite balanced between the perception of worsened (47%) and not worsened (53%) performance (Appendix A). Interestingly, using the univariate logistic regression models, we found almost two-thirds of the investigated socio-demographic and clinical variables were associated with the dichotomous outcome variable ‘worsened academic performance [yes vs. no]’. When applying the multivariable logistic regression model, the most important features directly associated with the perception of a negative impact of the pandemic on academic performance were: UCS total score, USS total score, introjected regulation, and boredom. An increased score in such variables was associated with a higher probability of perceiving a negative impact of the pandemic on overall academic performance (OR = 1.01, 1.05, 1.02, 1.37, respectively, see Table 4). Differently, a higher age and a higher score on pride were associated with a reduced probability of 7% and 32% (OR = 0.93 and OR = 0.68, respectively) of perceiving a negative impact of the pandemic on academic performance.

## 4. Discussion

This study aimed to evaluate the impact of the COVID-19 pandemic on the social relationships and academic lives of a sample of Italian university freshmen. To do this, the survey data collected during the year 2022 (post-pandemic) were compared to data collected in the pre-pandemic year, after the application of the propensity score matching. This procedure allowed us to highlight the psychological and clinical differences pre–post-pandemic aside from the potential socio-demographic differences and, in turn, to deduct that the differences may have been induced by the COVID-19 pandemic. The comparison made in this study found a lower, albeit still significant, total score on the GHQ-12 in the post-COVID-19 sample. This indicates that, even after the onset of the pandemic, freshmen continue to experience notable levels of mental distress. While the decrease in the total score might suggest a slight improvement, it is crucial to recognize that the overall value still falls within the range indicating significant mental distress. These findings corroborate the existing literature, which consistently highlights that freshmen are a particularly vulnerable population susceptible to elevated levels of psychological challenges and emotional strain during their transition to university life [4,41].

Results show that in the post-COVID-19 sample there was a decrease in substance use, such as alcohol, marijuana, and tobacco. Several factors may have contributed to this. Firstly, the COVID-19 pandemic and associated restrictions (i.e., social distancing measures, limitations on social gatherings) may have limited access to substances and reduced opportunities for peer influence [42]. The disruption in social contexts and changes in social dynamics during the pandemic may have led to a decrease in substance use among college students. Additionally, the increased focus on health and well-being during the pandemic may have influenced attitudes and behaviors related to a healthier lifestyle [43,44]. Heightened awareness of the potential health risks associated with substance use, the importance of maintaining a strong immune system, and the promotion of healthier lifestyles may have contributed to a decrease in substance use among university freshmen. At the same time, with the reduction in substance use assessed by the ASSIST, there was in the post-COVID-19 sample an increase in substance use as a coping strategy (Brief COPE), which can be explained as students needing to vent and feel less worried/fearful/angry/stressed or lonely [45].

Concerning the coping strategies, the results show that there has been an increase in the use of active coping among post-COVID-19 freshmen. This result can be attributed to several factors. The pandemic has brought about significant disruptions and challenges, such as remote learning, social isolation, and health concerns [46,47]. These unprecedented circumstances may have prompted students to adopt more active and problem-focused strategies to cope with the stressors associated with the pandemic [48]. Furthermore, the unpredictability of the pandemic may have increased students’ recognition of the importance of proactive coping [49]. Active coping strategies empower individuals to take control and directly address challenges, which may have become even more crucial in light of the uncertainties surrounding the pandemic and its impact on several aspects of life.

Another result concerns the decrease in the use of the planning coping strategy among post-COVID-19 freshmen. The pandemic has caused significant disruptions and uncertainties in various aspects of life [46,47]. The unpredictable nature of the pandemic and the need for immediate adaptation may have shifted students’ focus towards more immediate coping strategies rather than long-term planning and problem-solving. The pandemic-related restrictions, such as social distancing measures and remote learning, could have restricted students’ opportunities for future-oriented planning. The lack of certainty and control over the future, combined with the daily challenges posed by the pandemic, may have prompted freshmen to prioritize coping strategies that address immediate stressors rather than engaging in extensive planning. The psychological impact of the pandemic might have influenced students’ cognitive processes and prevented their ability to engage in effective planning [50]. The emotional burden during the pandemic may have redirected students’ coping efforts toward emotion-focused strategies rather than future-oriented problem-solving.

Post-COVID-19 freshmen also show a decrease in the acceptance coping strategy. During the pandemic, students may have experienced heightened difficulty in accepting and adapting to the uncertainties and changes imposed by the pandemic, which could have influenced their coping strategies [51]. Moreover, the social and physical distancing measures implemented during the pandemic may have limited freshmen social support networks and reduced opportunities for face-to-face interactions [52]. Social support is known to be an important resource for accepting and coping with difficult situations [53]. The reduction in available social support may have hindered the utilization of acceptance-based coping strategies among post-COVID-19 university freshmen.

Several factors may contribute to explaining the observed decrease in the use of the self-blame coping strategy. For instance, the pandemic and its associated challenges, such as academic disruptions, social isolation, and increased stressors, may have needed a shift towards more adaptive and effective coping strategies. Students may have recognized the limitations and negative consequences of self-blame as a coping strategy and sought alternative approaches. The increased awareness and availability of mental health support during the pandemic in universities [54] may have facilitated the adoption of more constructive coping strategies among university freshmen. The provision of online counseling services, mental health resources, and peer support networks could have promoted the exploration and utilization of healthier coping mechanisms, leading to a decrease in self-blame. Furthermore, the emphasis on resilience-building and psychological well-being in educational institutions during the pandemic may have influenced university freshmen to adopt coping strategies that promote self-compassion and self-care [55,56]. The recognition of the importance of self-acceptance and seeking external support may have contributed to the reduction in the use of self-blame as a coping mechanism.

The findings of this study suggest a noteworthy relationship between the perception of high levels of stress, as measured by the USS, and increased perceptions of the pandemic’s impact on social relationships among freshmen. The transition to university life is inherently stressful, and the added stressors brought on by the pandemic, such as remote learning, reduced social interactions, and uncertainty, may have compounded these stress levels [52,57]. Consequently, freshmen who have a higher perception of stress may be more inclined to perceive the pandemic as having a detrimental effect on their social relationships.

The results show a significant gender disparity in freshmen’s perception of the pandemic’s impact on social relationships: females reported a greater impact than males. This result confirms what was found in another Italian study, that females had a worse psychological condition than males [17]. This may be due to gender differences in coping strategies and social support utilization [58]. Females tend to rely more heavily on interpersonal relationships for emotional support and social connectedness [59]. The disruptions caused by the pandemic, such as physical distancing measures and increased isolation, may have had a more pronounced effect on females’ social relationships, leading to a heightened perception of the pandemic’s impact. Additionally, societal gender norms and expectations surrounding females’ roles as caregivers and nurturers may have influenced their heightened perception of the pandemic’s effect on their social relationships [60].

The results show that Italian freshmen reported a greater impact on their social relationships than foreign freshmen. Italians, being immersed in the local culture and societal expectations, may have internalized a stronger sense of collective responsibility, which, in turn, could have contributed to heightened disruptions in their social relationships [61,62]. Conversely, non-Italian freshmen may have maintained connections within their own cultural communities, experienced different cultural norms surrounding socialization, or relied on alternative means of social interaction, leading to a diminished perception of the pandemic’s influence.

Moreover, the more freshmen used instrumental support as a coping strategy, the greater the impact of the pandemic on their social relationships. This may be because instrumental support tends to be more strongly associated with problem-focused rather than emotion-focused coping. So, it is probable that those who have not been able to manage the strong emotions related to the pandemic have also suffered more in terms of the detriment of social relationships. Similarly, the more freshmen had external regulation, the greater the impact of the pandemic on their social life. People with this type of regulation are in fact motivated by external elements: control comes from the outside and not from individuals [63]. It is therefore probable that, for them, the pandemic had a greater impact on social relationships, linked to an absence of internal control over the events of their life.

Concerning the effect of the pandemic on academic performance, the sample was equally divided between freshmen who said that the pandemic has improved academic performance and freshmen who said it has worsened it. On the other hand, 60% of the freshmen said they had an improvement in their grades. This result confirms other research in which the academic performance of college students unexpectedly improved during the online learning period [64]. In contrast, an Italian study showed that only 30% of the students declared an improvement in their grade point average [7]. In particular, freshmen who were older and expressed more pride as a learning-related emotion reported a reduced likelihood of perceiving a negative impact of the pandemic on their grades. This result suggests that emotional experiences are important and could have made students more resilient toward COVID-19-related challenges and helped them learn more effectively online. On the contrary, those who reported more distress and the emotion of boredom linked to studying said that the pandemic had had a negative impact on their performance.

This study has a few limitations, primarily concerning the generalizability of the results, as the data were collected from a sample consisting of freshmen from a single university. Additionally, the use of web-based surveys may exclude students who are not digitally connected, potentially leading to a lack of representation for certain social groups with distinct socio-economic characteristics and lifestyles. Consequently, conducting an exclusively online survey may introduce significant bias and under-represent certain segments of the population.

## 5. Conclusions

This is one of the few studies that have investigated changes in the personal, social, and academic lives of freshmen before and after the COVID-19 pandemic. The results show that freshmen recruited after the pandemic had less psychological distress and substance use, indicative of better psychological well-being. Moreover, these freshmen used coping strategies that addressed immediate stressors rather than engaging in extensive planning. It is possible that the psychological effect of the pandemic has led freshmen’s coping efforts toward emotion-focused strategies rather than future-oriented problem-solving. This result highlights the importance of implementing interventions to mitigate the emotional impact of the pandemic to help freshmen regain perspective for the future. The pandemic seems to have impacted academic performance less and social relationships more, especially for females and Italian freshmen. Specific supportive interventions should therefore be developed for these target students. Further studies are needed to understand the long-term effects of the COVID-19 pandemic on university students, in order to identify the most effective strategies to support their well-being over time.

## Figures and Tables

**Table 1 ijerph-20-06485-t001:** Descriptive statistics and comparison of the socio-demographic and clinical features between the two survey samples matched by propensity score procedure.

	Survey 2019	Survey 2022	*p* Value ^$^
	n = 553	n = 553	
Gender			
Male	234 (42.3%)	245 (44.3%)	0.800
Female	317 (57.3%)	306 (55.3%)	
Other	2 (0.4%)	2 (0.4%)	
Age			
Mean (SD)	22.9 (3.80)	20.9 (3.81)	<0.001 (t)
Nationality			
Italian	527 (95.3%)	521 (94.2%)	0.500
Other	26 (4.7%)	32 (5.8%)	
Marital status			
Single	257 (46.5%)	272 (49.2%)	0.399
Relationship	296 (53.5%)	281 (50.8%)	
University status			
Student	424 (76.7%)	407 (73.6%)	0.266
Worker	129 (23.3%)	146 (26.4%)	
Living status			
In town	344 (62.2%)	362 (65.5%)	0.287
Out-town	209 (37.8%)	191 (34.5%)	
UCS Total score			
Mean (SD)	81.4 (17.6)	83.4 (17.2)	0.113
GHQ-12 Total			
Mean (SD)	6.44 (3.00)	5.96 (2.90)	0.007
P4 Screener (suicide risk)			
No	463 (83.7%)	473 (85.5%)	0.453
Yes	90 (16.3%)	80 (14.5%)	
USS Total score			
Mean (SD)	14.3 (7.54)	14.1 (8.39)	0.285
Brief COPE Active coping			
Mean (SD)	5.29 (1.39)	5.57 (1.41)	<0.001
Brief COPE Planning			
Mean (SD)	5.97 (1.47)	5.66 (1.54)	0.001
Brief COPE Positive reframing			
Mean (SD)	4.81 (1.61)	4.77 (1.51)	0.841
Brief COPE Acceptance			
Mean (SD)	5.60 (1.42)	5.42 (1.46)	0.027
Brief COPE Humor			
Mean (SD)	3.82 (1.54)	3.96 (1.52)	0.060
Brief COPE Religion			
Mean (SD)	3.03 (1.56)	3.02 (1.52)	0.999
Brief COPE Emotional support			
Mean (SD)	4.66 (1.71)	4.64 (1.71)	0.939
Brief COPE Instrumental support			
Mean (SD)	4.78 (1.64)	4.77 (1.67)	0.815
Brief COPE Self- distraction			
Mean (SD)	5.07 (1.46)	4.96 (1.45)	0.317
Brief COPE Denial			
Mean (SD)	2.75 (1.13)	2.83 (1.17)	0.261
Brief COPE Venting			
Mean (SD)	4.48 (1.61)	4.38 (1.57)	0.290
Brief COPE Substance use			
Mean (SD)	2.37 (1.01)	2.44 (0.993)	0.015
Brief COPE Behavioral disengagement			
Mean (SD)	3.12 (1.28)	3.26 (1.39)	0.156
Brief COPE Self-blame			
Mean (SD)	5.84 (1.49)	5.63 (1.55)	0.036
ASSIST Total Tobacco			
Mean (SD)	5.16 (7.28)	3.57 (7.12)	<0.001
ASSIST Total Alcohol			
Mean (SD)	7.11 (5.97)	4.10 (4.23)	<0.001
ASSIST Total Marijuana			
Mean (SD)	1.76 (4.61)	0.642 (2.73)	<0.001
ASSIST Total Cocaine			
Mean (SD)	0.0850 (0.831)	0.0217 (0.434)	0.056
ASSIST Total Stimulants			
Mean (SD)	0.0325 (0.360)	0.0416 (0.670)	0.480
ASSIST Total Inhalants			
Mean (SD)	0.0217 (0.312)	0.0380 (0.665)	0.999
ASSIST Total Sedatives			
Mean (SD)	0.544 (3.07)	0.394 (2.40)	0.164
ASSIST Total Hallucinogens			
Mean (SD)	0.0488 (0.458)	0.0380 (0.614)	0.205
ASSIST Total Opioids			
Mean (SD)	0.0452 (0.876)	0 (0)	0.157
ASSIST Total Other drugs			
Mean (SD)	0.0217 (0.312)	0.0181 (0.282)	0.998

Note. ^$^ Mann–Whitney test or *t*-test where indicated with (t) and Chi-squared test for categorical variables.

**Table 2 ijerph-20-06485-t002:** Univariate logistic regression model output for assessing the association between the socio-demographic, clinical variables and the dichotomous dependent variable ‘effect of pandemic on social relationships’.

	Odd Ratio	95% CI		*p* Value
Variables		Lower Bound	Upper Bound	
Age	0.96	0.92	1.01	0.102
Gender (female vs. male *)	1.61	1.08	2.42	**0.021**
Nationality (Italian vs. other *)	0.34	0.17	0.72	**0.004**
Marital Status (single vs. in relationship *)	1.13	0.76	1.69	0.540
University Status (student vs. worker *)	1.30	0.82	2.11	0.274
Living Status (in town vs. out-town *)	0.99	0.65	1.52	0.971
UCS Total score	0.99	0.98	1.01	0.295
GHQ-12 Total score	1.07	1.00	1.15	0.062
P4 Screener (suicide risk vs. no risk *)	0.78	0.46	1.37	0.376
USS Total score	1.04	1.02	1.07	**0.002**
Brief COPE Active coping	1.08	0.94	1.24	0.299
Brief COPE Planning	1.13	0.99	1.28	0.071
Brief COPE Positive reframing	1.03	0.90	1.18	0.669
Brief COPE Acceptance	1.01	0.88	1.16	0.863
Brief COPE Humor	0.98	0.85	1.11	0.722
Brief COPE Religion	1.11	0.97	1.28	0.151
Brief COPE Emotional support	1.19	1.06	1.35	**0.005**
Brief COPE Instrumental support	1.20	1.07	1.37	**0.003**
Brief COPE Self-distraction	0.98	0.85	1.12	0.740
Brief COPE Denial	1.10	0.93	1.33	0.290
Brief COPE Venting	1.16	1.02	1.32	**0.030**
Brief COPE Substance use	0.94	0.78	1.15	0.549
Brief COPE Behavioral disengagement	1.03	0.89	1.19	0.740
Brief COPE Self-blame	1.03	0.90	1.17	0.654
ASSIST Total Tobacco	1.02	0.99	1.05	0.317
ASSIST Total Alcohol	1.01	0.97	1.07	0.620
ASSIST Total Marijuana	1.01	0.94	1.11	0.748
ASSIST Total Cocaine **^#^**	614.00	0.00	--	0.985
ASSIST Total Stimulants	0.85	0.58	1.12	0.253
ASSIS Total Inhalants	1.02	0.77	1.91	0.914
ASSIST Total Sedatives	1.02	0.94	1.14	0.707
ASSIST Total Hallucinogens **^#^**	445.50	0.00	--	0.981
ASSIST Total Opioids **^#^**	1.11	--	--	--
ASSIST Total Other drugs	0.68	0.27	1.30	0.254
SRK Knowledge extraction	1.06	0.98	1.13	0.132
SRK Knowledge networking	1.05	0.98	1.13	0.137
SRK Knowledge practice	1.01	0.93	1.10	0.809
SRK Knowledge critique	1.03	0.96	1.11	0.370
SRK Knowledge monitoring	1.01	0.92	1.10	0.882
AMS Lack of motivation	1.03	1.00	1.06	**0.042**
AMS External regulation	1.03	1.01	1.06	**0.020**
AMS Introjected regulation	1.02	1.00	1.04	0.076
AMS Identified regulation	1.01	0.99	1.03	0.244
AMS Intrinsic regulation	1.00	0.97	1.02	0.898
SASP Total score	0.99	0.95	1.02	0.425
IAUQ Total score	1.02	0.99	1.04	0.139
Enjoyment	1.01	0.77	1.32	0.946
Hope	0.80	0.63	1.00	0.053
Pride	0.75	0.58	0.96	**0.022**
Anger	1.10	0.88	1.38	0.416
Anxiety	1.19	0.95	1.49	0.130
Shame	1.28	1.05	1.56	**0.014**
Hopelessness	1.34	1.10	1.65	**0.005**
Boredom	1.24	1.00	1.55	0.054
SOGS Total score	0.94	0.77	1.19	0.561
Multivariable model including all significant variables above (Estimation carried out by STEPwise variable selection model)				
Gender (female vs. male *)	1.69	1.09	2.62	0.018
Nationality (Italian vs. other *)	0.24	0.11	0.54	<0.001
USS Total score	1.04	1.01	1.07	0.022
Brief COPE Instrumental Support	1.17	1.03	1.34	0.018
AMS External regulation	1.03	1.002	1.06	0.046

Note. ^#^ Estimate not reliable due to missing data; in bold significant *p* values < 0.05; underlined *p* values < 0.1. * Ref: reference category.

**Table 3 ijerph-20-06485-t003:** Univariate logistic regression model output for assessing the association between the socio-demographic, clinic variables and the dichotomous dependent variable ‘effect of pandemic on academic grade’.

	Odd Ratio	95% CI		*p* Value
Variables		Lower Bound	Upper Bound	
Age	0.99	0.94	1.03	0.530
Gender (female vs. male *)	0.90	0.64	0.92	0.565
Nationality (Italian vs. other *)	1.02	0.48	2.10	0.954
Marital status (single vs. in relationship *)	1.04	0.74	1.46	0.836
University status (student vs. worker *)	0.98	0.66	1.43	0.904
Living status (in town vs. out-town *)	1.23	0.86	1.76	0.249
UCS Total score	0.99	0.98	1.00	0.188
GHQ-12 Total score	0.96	0.91	1.02	0.186
P4 Screener (suicide risk vs. no risk *)	1.34	0.83	2.16	0.229
USS Total score	1.02	1.00	1.04	0.132
Brief COPE Active coping	0.88	0.78	1.00	**0.043**
Brief COPE Planning	0.98	0.88	1.10	0.763
Brief COPE Positive reframing	0.91	0.82	1.02	0.125
Brief COPE Acceptance	0.93	0.83	1.05	0.236
Brief COPE Humor	0.89	0.80	1.00	**0.049**
Brief COPE Religion	0.98	0.87	1.09	0.714
Brief COPE Emotional support	1.08	0.98	1.19	0.134
Brief COPE Instrumental support	1.06	0.96	1.18	0.233
Brief COPE Self-distraction	1.07	0.95	1.20	0.270
Brief COPE Denial	1.14	0.98	1.31	0.084
Brief COPE Venting	1.15	1.03	1.29	**0.011**
Brief COPE Substance use	1.18	1.00	1.40	0.054
Brief COPE Behavioral disengagement	1.11	0.99	1.26	0.082
Brief COPE Self-blame	1.04	0.93	1.16	0.477
ASSIST Total Tobacco	0.99	0.97	1.02	0.478
ASSIST Total Alcohol	1.00	0.96	1.05	0.829
ASSIST Total Marijuana	0.97	0.90	1.04	0.422
ASSIST Total Cocaine	1.29	0.83	4.76	0.412
ASSIST Total Stimulants	1.39	0.96	5.09	0.313
ASSIST Total Inhalants **^#^**	902.02	0.00	--	0.976
ASSIST Total Sedatives	1.17	1.06	1.35	**0.007**
ASSIST Total Hallucinogens	1.09	0.81	1.64	0.538
ASSIST Total Opioids **^#^**	1.09	--	--	--
ASSIST Total Other drugs	1.20	0.63	2.81	0.558
SRK Knowledge extraction	0.99	0.93	1.05	0.786
SRK Knowledge networking	0.95	0.89	1.01	0.079
SRK Knowledge practice	0.94	0.87	1.01	0.097
SRK Knowledge critique	0.96	0.91	1.02	0.234
SRK Knowledge monitoring	0.91	0.84	0.98	**0.020**
AMS Lack of motivation	1.02	1.00	1.04	0.064
AMS External regulation	1.01	0.99	1.03	0.297
AMS Introjected regulation	0.99	0.97	1.01	0.277
AMS Identified regulation	0.98	0.97	1.00	0.062
AMS Intrinsic regulation	0.98	0.96	1.00	**0.028**
SASP Total score	1.01	0.98	1.04	0.499
IAUQ Total score	1.02	1.00	1.04	**0.018**
Enjoyment	1.10	0.87	1.40	0.412
Hope	1.12	0.93	1.36	0.243
Pride	0.97	0.80	1.19	0.804
Anger	1.00	0.83	1.21	0.985
Anxiety	0.87	0.72	1.06	0.168
Shame	0.99	0.85	1.16	0.915
Hopelessness	1.06	0.90	1.25	0.463
Boredom	1.03	0.86	1.24	0.715
SOGS Total score	1.28	1.04	1.64	**0.032**
Multivariable model including all significant variables above (Estimation carried out by STEPwise variable selection method)				
Brief COPE Venting	1.15	1.03	1.29	**0.012**
ASSIST Total Sedatives	1.15	1.05	1.31	**0.014**
AMS Intrinsic regulation	0.98	0.96	0.99	**0.046**
SOGS Total	1.24	1.01	1.60	**0.053**

Note. ^#^ Estimate not reliable due to missing data; in bold significant *p* values < 0.05; underlined *p* values < 0.1. * Ref: reference category.

**Table 4 ijerph-20-06485-t004:** Univariate logistic regression model output for assessing the association between the socio-demographic, clinic variables and the dichotomous dependent variable ‘effect of pandemic on academic performance’.

	Odd Ratio	95% CI		*p* Value
Variables		Lower Bound	Upper Bound	
Age	0.93	0.87	0.98	**0.008**
Gender (female vs. male *)	1.05	0.75	1.12	0.782
Nationality (Italian vs. other *)	0.66	0.31	1.36	0.269
Marital status (single vs. in relationship *)	1.03	0.73	1.43	0.880
University status (student vs. worker *)	1.13	0.78	1.66	0.517
Living status (in town vs. out-town *)	0.80	0.56	1.14	0.223
UCS Total score	0.99	0.98	1.00	**0.004**
GHQ-12 Total score	1.14	1.07	1.21	**0.000**
P4 Screener (suicide risk vs. no risk *)	1.45	0.90	2.35	0.123
USS Total score	1.07	1.04	1.09	**0.000**
Brief COPE Active coping	0.84	0.74	0.95	**0.004**
Brief COPE Planning	0.84	0.75	0.94	**0.002**
Brief COPE Positive reframing	0.96	0.86	1.07	0.468
Brief COPE Acceptance	0.91	0.81	1.03	0.129
Brief COPE Humor	0.95	0.85	1.06	0.372
Brief COPE Religion	1.13	1.01	1.26	**0.035**
Brief COPE Emotional support	1.12	1.02	1.24	**0.022**
Brief COPE Instrumental support	1.09	0.99	1.21	0.086
Brief COPE Self-distraction	1.06	0.95	1.20	0.293
Brief COPE Denial	1.32	1.14	1.53	**0.000**
Brief COPE Venting	1.10	0.99	1.22	0.082
Brief COPE Substance use	1.35	1.13	1.63	**0.001**
Brief COPE Behavioral disengagement	1.30	1.15	1.48	**0.000**
Brief COPE Self-blame	1.04	0.93	1.16	0.514
ASSIST Total Tobacco	1.02	0.99	1.04	0.163
ASSIST Total Alcohol	1.04	1.00	1.09	**0.049**
ASSIST Total Marijuana	1.06	1.00	1.15	0.080
ASSIST Total Cocaine	1.24	0.80	4.42	0.469
ASSIST Total Stimulants	0.94	0.62	1.22	0.637
ASSIST Total Inhalants	1.10	0.83	1.81	0.540
ASSIST Total Sedatives	1.07	0.99	1.18	0.099
ASSIST Total Hallucinogens	1.06	0.79	1.58	0.671
ASSIST Total Opioids **^#^**	1.09	--	--	--
ASSIST Total Other drugs **^#^**	0.00	--	--	0.978
SRK Knowledge extraction	1.00	0.94	1.06	0.983
SRK Knowledge networking	1.00	0.94	1.06	0.962
SRK Knowledge practice	0.92	0.85	0.98	**0.016**
SRK Knowledge critique	1.06	1.00	1.12	0.060
SRK Knowledge monitoring	0.88	0.81	0.95	**0.002**
AMS Lack of motivation	1.05	1.03	1.07	**0.000**
AMS External regulation	1.05	1.03	1.07	**0.000**
AMS Introjected regulation	1.03	1.01	1.04	**0.002**
AMS Identified regulation	0.99	0.97	1.00	0.113
AMS Intrinsic regulation	0.98	0.96	1.00	**0.020**
SASP Total score	0.94	0.92	0.97	**0.000**
IAUQ Total score	1.03	1.01	1.05	**0.002**
Enjoyment	0.63	0.50	0.80	**0.000**
Hope	0.60	0.49	0.73	**0.000**
Pride	0.55	0.44	0.68	**0.000**
Anger	1.36	1.12	1.65	**0.002**
Anxiety	1.39	1.14	1.68	**0.001**
Shame	1.54	1.31	1.82	**0.000**
Hopelessness	1.64	1.39	1.95	**0.000**
Boredom	1.67	1.39	2.03	**0.000**
SOGS Total score	1.02	0.84	1.24	0.852
Multivariable model including all significant variables above (Estimation carried out by STEPwise variable selection method)				
Age	0.93	0.87	0.99	**0.024**
UCS Total score	1.01	1.002	1.03	**0.025**
USS Total score	1.05	1.02	1.08	**<0.001**
AMS Introjected regulation	1.02	1.003	1.04	**0.024**
Pride	0.68	0.52	0.88	**0.004**
Boredom	1.37	1.09	1.73	**0.007**

Note. ^#^ Estimate not reliable due to missing data; in bold significant *p* values < 0.05; underlined *p* values < 0.1. * Ref: reference category.

## Data Availability

The data that support the findings of this study are available on request from the corresponding author.

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
