# Peer review of "Impact of COVID-19 Pandemic on Well-Being, Social Relationships and Academic Performance in a Sample of University Freshmen: A Propensity Score Match Evaluation Pre- and Post-Pandemic"

_ijerph, 2023, doi:10.3390/ijerph20156485_

Round 1

Reviewer 1 Report

Dear authors, here are some suggestions to improve your article.

 Regarding the introduction:

The article is relevant because of the topic being worked on. In addition, it is necessary to investigate in university population the influence that the pandemic has had on students. Although distance education is cited, references are missing.

I recommend reading these articles:

Abreu, J. L. (2020). Times of Coronavirus: Online Education in Response to the Crisis. Int. J. Good Consci. 15, 1–15.

Alemany-Arrebola I, Rojas-Ruiz G, Granda-Vera J and Mingorance-Estrada ÁC (2020). Influence of COVID-19 on the Perception of Academic Self-Efficacy, State Anxiety, and Trait Anxiety in College Students. Front. Psychol. 11:570017. doi: 10.3389/fpsyg.2020.570017

Kelly, K. (2020). Results from Top Hat’s COVID-19 Student Survey about Online Learning PhilonEdTech. Available online at: https://url2.cl/uFZiv

Marinoni, G., Van’t Land, H., and Jensen, T. (2020). The impact of COVID-19 on Higher Education around the world. IAU Global Survey Report UNESCO.

Regarding Materials and Methods.

In the Design of the study I do not consider this study as longitudinal, because although it is measured in two different time periods, it is not the same sample, they are two different samples of first year.

In the section on instruments, the reading is complicated because there are many tests administered. It would be advisable to include a table specifying the scales, the forms of measurement, the type of response and the reliability of the tests, which is not always clearly specified in the text.

 In the "Results" section, when analyzing so many scales, it is necessary to return to the instruments section to recall the test analyzed. This makes reading difficult.

It is not specified how and when performance is measured and in which subject. In addition, it is not specified whether there are students who drop out.

 In relation to the discussion, distance learning is cited, but in the introduction there is no reference to how the methodology used may influence performance and the emotional factors.

 I recommend:

 1. Include some recommended references.

2. Design a table with the instruments to make it easier to read.

3. Specify how the student's academic performance is measured and in which subject.

Author Response

Dear reviewer,

Thanks and best regards

Reviewer 2 Report

The authors have conveyed a well-thought-of study into an interesting paper. However, here are some points of improvement.

1. The title can be changed to reflect the main and novel findings of the study. (Line 2-3)

2. The abstract did not specify the cohort demographics and sample size. Likewise, it would be an easier read if the type of study can be introduced in the title or the abstract. In addition, the abstract mentioned "longitudinal study". Is this implying that the current study is one? If so, there might be a conflict in the sample selection, as the authors explicitly mentioned that there are two samples. Longitudinal studies are done using the sample over time.

3. In the introduction section, being in the first year of college is conveyed as a stressor, but there were no succeeding explanations after that. There should be a separate discussion on the first year of college and how it invokes stress, and COVID-19's effect on stress. (Line 36-38)

4. In the materials and methods section, the authors could specify a reporting guideline used.

5. Clear inclusion and exclusion criteria should also be included.

6. The Brief COPE Inventory yielded a below-average reliability scale of 50%. How was the data treated? (Line 124)

7. The evaluation of the effect o the pandemic was presumed to be carried out using a researcher-made questionnaire. Was validity testing done? (Line 186-187)
8. In Line 231, there is a mention of "after COVID-19". As per current literature, the WHO might not have clearly stated an end to the pandemic yet (despite an earlier declaration this year).

9. The tables can be trimmed neatly to be more presentable.

10. As mentioned earlier, the authors need to prove through literature that the post-pandemic era has started, to rationalize the comparison between the two cohorts in this study.

11. In Line 404, the authors should use the word "perceptions" to highlight the limitation of using questionnaires without a biomarker for stress, or actual performance ratings in school.

12. The conclusion did not give a summary of the findings and/or highlight the novelty of the study. (Line 460-471)

13. There are some papers that the authors may have missed. It would be great to include these and establish the gap in the literature that the authors want to address.

https://www.ncbi.nlm.nih.gov/pmc/articles/PMC8469053/

https://bmcpsychology.biomedcentral.com/articles/10.1186/s40359-021-00649-9

https://link.springer.com/article/10.1007/s10804-023-09444-9

Author Response

Dear reviewer,

Thanks and best regards

Round 2

Reviewer 2 Report

The authors have made considerable changes to their manuscript. Thank you very much.